# Melt in Antarctica derived from SMOS observations at L band

Marion Leduc-Leballeur[1,2], Ghislain Picard[2], Giovanni Macelloni[1], Arnaud Mialon[3], and Yann K. Kerr[3]

[1]Institute of Applied Physics "Nello Carrara" – National Council of Research, 50019 Sesto Fiorentino, Italy
[2]UGA, CNRS, Institut des Géosciences de l'Environnement (IGE), UMR 5001, Grenoble, 38041, France
[3]CESBIO (CNES, CNRS, IRD, UPS), Univ. Toulouse, 31401 Toulouse Cedex 09, France

**Correspondence:** Leduc-Leballeur (m.leduc@ifac.cnr.it)

**Abstract.** Melt occurrence in Antarctica is derived from L-band observations from the Soil Moisture and Ocean Salinity (SMOS) satellite between the austral summer 2010/11 and 2017/18. The detection algorithm is adapted from a threshold method previously developed for 19 GHz passive microwave measurements from Special Sensor Microwave Imagers (SSM/I, SSMIS). The comparison of daily melt occurrence retrieved from 1.4 GHz and 19 GHz observations shows an overall close agreement, but a lag of few days is usually observed by SMOS at the beginning of the melt season. To understand the difference, a theoretical analysis is performed using a microwave emission radiative transfer model and shows that the sensitivity of 1.4 GHz signal to liquid water is significantly weaker than at 19 GHz if the water is only present in the uppermost tens of centimeters of the snowpack. Conversely, 1.4 GHz measurements are sensitive to water when spread over at least 1 m and when present at depth, up to hundreds of meters. This is explained by the large penetration depth in dry snow and by the long wavelength (21 cm). We conclude that SMOS and higher frequency radiometers provide interesting complementary information on melt occurrence and on the location of the water in the snowpack.

## 1 Introduction

Melt occurs in coastal Antarctica and on ice shelves during the austral summer. Its duration and extent are useful climate indicators due to their connection to surface temperature and surface energy budget (e.g. Liu et al., 2006; Picard et al., 2007). Moreover, intense melting event has been identified as a precursor of some major ice shelf collapses (Scambos et al., 2000). Thus, monitoring of the melt season contributes to characterize the seasonal and inter-annual climatic variations in Antarctica and is important to assess the future stability of the ice-sheet (Golledge et al., 2015).

Remote sensing offers a particularly relevant means to obtain information over the entire Antarctic continent and over long-term periods, given the very rare in situ measurements related to melt or liquid water (Jakobs et al., 2019). Microwave frequencies have been widely used to detect melt in polar regions exploiting the marked variation of the signal due to the high absorption of microwaves by water relative to that of dry snow. Various detection algorithms have been developed for active sensors (e.g. Nghiem et al., 2001, 2005; Ashcraft and Long, 2006; Kunz and Long, 2006; Hall et al., 2009; Trusel et al., 2012; Zheng et al., 2019) and passive sensors (e.g. Mote et al., 1993; Ridley, 1993; Zwally and Fiegles, 1994; Abdalati and Steffen, 1997; Torinesi et al., 2003; Liu et al., 2005, 2006; Tedesco, 2007; Tedesco et al., 2007) and applied to the Greenland and Antarctic ice sheets.

In the case of radiometer measurements, studies have mainly used 19 GHz and 37 GHz frequencies available since 1979 from several satellite sensors such as the Scanning Multichannel Microwave Radiometer (SMMR) on the Nimbus 7 satellite or the Special Sensor Microwave/Imager (SSM/I) and Special Sensor Microwave Imager Sounder (SSMIS) from the Defense Meteorological Satellite Program (DMSP) satellites. Since 2009, the Soil Moisture and Ocean Salinity (SMOS) satellite has provided radiometric observations at L band, a frequency capable of penetrating much deeper in the ice sheets, on the order of several hundred meters at 1.4 GHz (Passalacqua et al., 2018) compared to only a few meters for the higher frequencies (Surdyk, 2002). This suggests that L-band observations could offer new information on melt.

The aim of this study is to retrieve melt in Antarctica from daily SMOS observations, and to investigate the similarities and differences with melt detected at 19 GHz. Section 2 introduces the data sets. Section 3 describes the method to detect melt and Section 4 compares the daily melt occurrence obtained with 1.4 GHz and 19 GHz observations. Section 5 presents a modeling study to assess the liquid water sensitivity of brightness temperature ($T_B$) at 1.4 GHz and to discuss the differences with 19 GHz.

## 2  Data sets

### 2.1  SMOS observations

The SMOS mission was developed by the European Space Agency (ESA) in collaboration with the Centre National d'Etudes Spatiales (CNES) in France and the Centro para el Desarrollo Tecnologico Industrial (CDTI) in Spain. This satellite is operated by CNES and ESA and carries on board a L-band interferometric radiometer operating at 1.4 GHz (21 cm) with an averaged ground resolution of 43 km (Kerr et al., 2010). The radiometer provides multi-angular fully polarized $T_B$ (Kerr et al., 2001).

The SMOS Level 3 product delivers multi-angular $T_B$ at top of the atmosphere in the antenna polarization reference frame (Al Bitar et al., 2017). The product is georeferenced on the Equal-Area Scalable Earth version 2.0 grid (EASE–Grid 2; Brodzik et al. (2012)), with an over-sampled resolution of about 628 km$^2$, which is distorted in the polar regions (around $100 \times 6$ km$^2$ as latitude$\times$longitude). It comprises daily-average and incident angle-average with angle bins every 5º from 0º to 65º. $T_B$ at vertical (V) and horizontal (H) polarizations for the 50-55º average range of incidence angle are used here. They come from the RE04 reprocessed version between April 2010 and April 2015, and from operational version between May 2015 to March 2018, both distributed by CATDS (Centre Aval de Traitement des Données SMOS; www.catds.fr).

The gaps shorter than 3 days in the SMOS time series are filled by a linear interpolation. Longer gaps result in missing values in the product. If more than 60 days are missing over a year, the grid point is ignored for that year (about 7% of pixel every year, mainly south of 83ºS).

The land-ocean mask used comes from the Land-Ocean-Coastline-Ice classification associated to the EASEGrid 2.0 map projections and derived from the MODIS land cover product by Brodzik and Knowles (2011) (available on https://nsidc.org/data/nsidc-0609).

## 2.2 Observations at 19 GHz and daily surface melting

Satellite observations at 19 GHz were acquired by the Special Sensor Microwave/Imager (SSM/I) and Special Sensor Microwave Imager Sounder (SSMIS), processed by the National Snow and Ice Data Center (NSIDC, Maslanik and Stroeve (2004, updated 2018)).

Daily $T_B$ observations at H polarization are processed according to Picard and Fily (2006) to derive daily surface melt from 1979 to 2018 (available on http://gp.snow-physics.science/melting). This data set provides daily melt status, i.e. presence or absence of liquid water, for every grid point on the Southern stereographic polar grid with a grid spacing of 25 km$^2$. The effective resolution of the product is coarser, of the order of 40 km, close to that provided by SMOS.

To compare SMOS and SSMIS data sets, the SSMIS observations and products are collocated within the SMOS grid using the nearest neighbour method. If the nearest neighbour is not flagged as "land" in the SSMIS grid, the pixel was removed from the analysis to avoid the error of comparison between the two frequencies. In this way, about 50 pixels are excluded, which doesn't affect the statistical significance of the comparison results.

## 3 Melting detection method

The algorithm to detect melt occurrence from the 1.4 GHz observations is inspired by the work at 19 GHz of Torinesi et al. (2003), itself based on Zwally and Fiegles (1994). The algorithm determines an optimal threshold for every year in every pixel, and considers that any daily $T_B$H over this threshold indicates melting occurrence. $T_B$ is measured at large observation angles (above 50$^o$). In this configuration, the H polarization is favored because the emissivity of dry firn is usually significantly lower at H than at V polarization, while the emissivity of wet firn is always close to 1 at both polarizations. It results that the increase in $T_B$ from dry to wet snow is more significant at H polarization, and easier to detect.

The algorithm uses an adaptive threshold $T$ in each grid point and for each year given by $T = M + a\sigma$, with $M$ the time average and $\sigma$ the standard deviation of $T_B$ when snow is dry. According to the analysis of daily air surface temperature, Torinesi et al. (2003) found a suitable value of $a = 3$ so that most melting events correspond to daily maximum temperatures above -5$^o$C. This value is also typical for outliers detection (e.g. von Storch and Zwiers, 2001).

To solve the circular problem of computing $M$ and $\sigma$ for non melting days in order to detect melting days, the initial step consists in calculating $M$ in each grid point on a fixed period of one year – from 1 April to 31 March – and in setting $a\sigma$ to a first-guess fixed value. Previous studies for 19 GHz used $a\sigma = 30$ K. However, we found it unsuitable at 1.4 GHz, because of the weaker sensitivity to liquid water (Section 5). We instead propose a lower first guess value of $a\sigma = 15$ K.

With these assumptions, a first guess melt time series is detected and new estimates of $M$ and $\sigma$ are computed by removing melting days from the $T_B$ series, still limiting the period from 1 April and 31 March. Melt is then detected once again using the updated threshold. The process is iterated three times to ensure stable estimates. The algorithm returns a binary indicator for each day and each grid point, 0 for the absence and 1 for the presence of liquid water.

This algorithm needs further correction for some false alarms found on the Antarctic Plateau where melt is known to never occur. These alarms are likely due to variations of $T_B$H of the order of several Kelvin that were reported by Brucker et al.

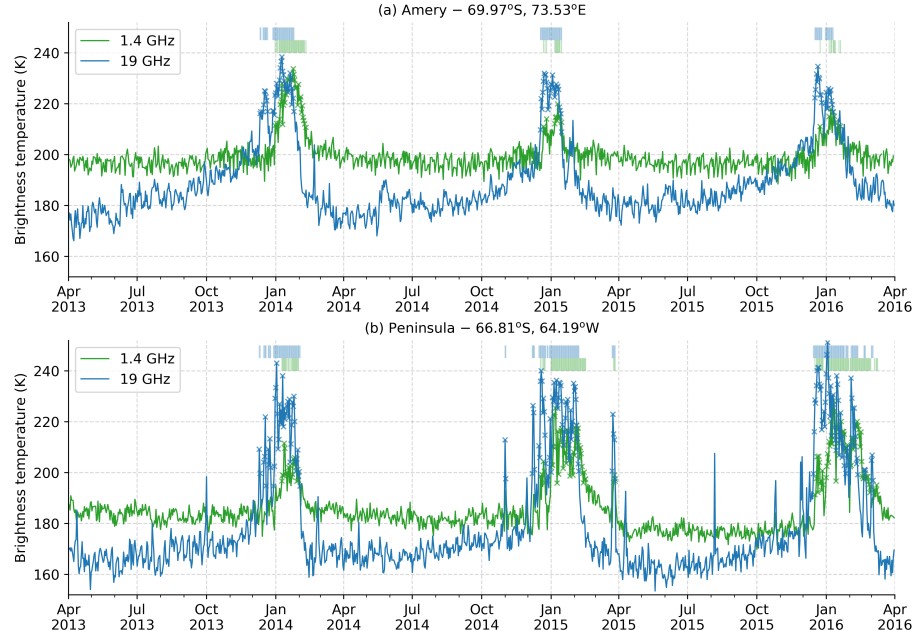

**Figure 1.** Brightness temperature at H polarization (K) at 1.4 GHz (green) and 19 GHz (blue) from April 2013 to March 2016 on (a) the Amery area and (b) the Antarctic Peninsula. The melting days detected by each frequency are depicted by crosses on the time series and recalled by pale lines above.

(2014) and Leduc-Leballeur et al. (2017) and explained to result from the snow metamorphism and surface hoar removal by wind storms. Noting that these changes do not impact $T_B V$, although melt does, we consider here that the areas with low annual standard deviation of $T_B V$ are not subject to melt. We estimated a threshold standard deviation of 2.8 K based on the fact that it excludes 95 % of grid points with surface elevation higher than 1500 m. Thus, as a final step of the algorithm, the grid points with a $T_B V$ annual standard deviation lower than this threshold are masked out that year.

## 4 Comparison with 19 GHz

Figure 1 shows two examples of two consecutive melt seasons in the Amery area (69.97°S, 73.53°E) and the Antarctic Peninsula (66.81°S, 64.19°W). For each event, melt is detected several days earlier at 19 GHz compared to 1.4 GHz. For instance, in 11 December 2013 on the Amery time series, a short melting event lasting 6 days is missed at 1.4 GHz while it is well detected at 19 GHz. This suggests that this event was weak and only affected the superficial part of the snowpack. On the other hand, the short melting event during March 2015 on the Peninsula time series is detected by both frequencies, suggesting intense melt with percolation in a large upper part of the snowpack.

The beginning of the melt season detected usually largely differs between both frequencies as illustrated in Figure 2. On average, the first melting day can be detected as early as September at 19 GHz, while it is rare to detect melt earlier than

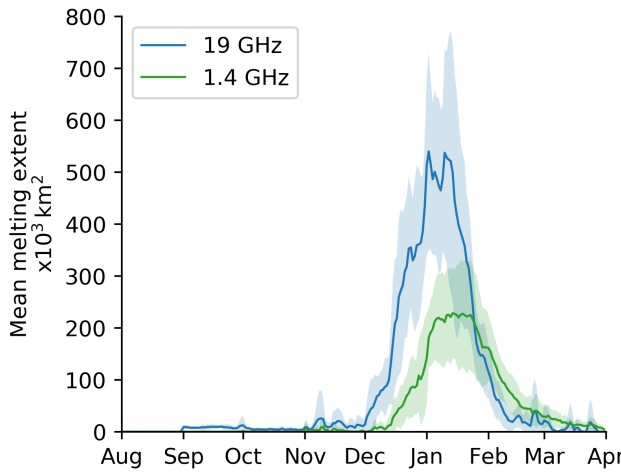

**Figure 2.** Daily mean melting extent from April 2010 to March 2018 detected with observations at 1.4 GHz (green) and at 19 GHz (blue). Standard deviation is in pale area.

December at 1.4 GHz. For the pixel where melt is detected by both frequencies in a given year, the 19 GHz detection precedes by 1-5 days for 28% of the pixels and by 6-15 days for 26% of them. This lag is also observed for the end of the season with a persistence of the melt detected at 1.4 GHz until nearly April.

Figure 2 also highlights that the melt extent detected at 19 GHz is 3 to 6 times as large as at 1.4 GHz, depending on the years. The standard deviation maximum is reached in January at 250,000 km$^2$ and 110,000 km$^2$ for 19 GHz and 1.4 GHz, respectively. Spatial variations are illustrated by Figure 3, which shows the annual mean duration of melt season between April 2010 and March 2018 detected at both frequencies. Melting is concentrated on the coast with a maximum in the Antarctic Peninsula as previously reported for 19 GHz (Tedesco, 2009; Kuipers Munneke et al., 2012; Datta et al., 2018, 2019; Scott et al., 2019). The largest differences are observed in Filchner and Ross ice shelves where melt is detected to occur a few days every year at 19 GHz, but is insufficient to be detected at 1.4 GHz. The difference is certainly explained by the difference of sensitivity. Indeed, as these ice shelves only experience limited melt, the liquid water is likely concentrated in the uppermost few centimeters of the snowpack.

Figure 3 and 4 highlight that 19 GHz is more effective to detect short melting duration than 1.4 GHz. Indeed, more than 55% of the pixels where melt occurs remain wet for less than 10 days in a year according to 19 GHz observations, and about 20% remain wet between 11 and 20 days. At 1.4 GHz, the duration of the melt season is usually longer, in only 20% of the pixels subject to melt, the season is 1-10 days, it is 11-40 days in 55% of the pixels. This hints that SMOS is only sensitive to long and intense melt seasons.

However, it also happens that some melting days are detected with the 1.4 GHz observations but not with the 19 GHz observations. This case is illustrated with the example of the Antarctic Peninsula provided by Figure 5 for the three summer seasons from 2013 to 2016. This area is known to be subjected each year to a long melting season, but an high interannual

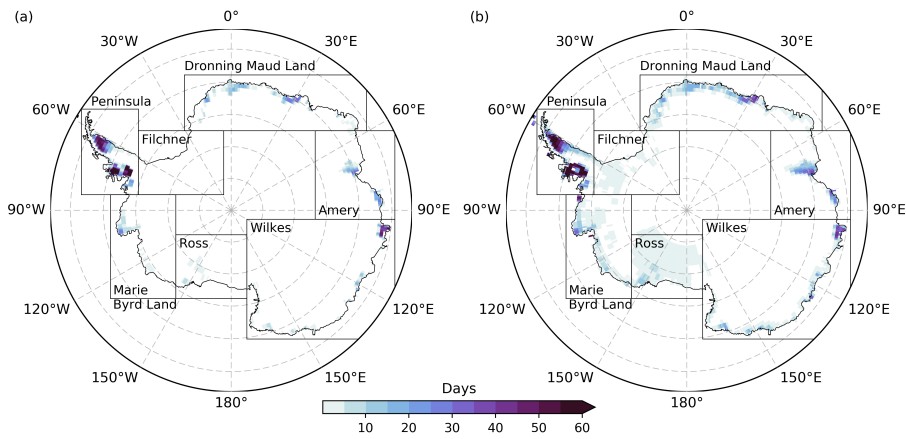

**Figure 3.** Annual mean of melting duration (days) from April 2010 to March 2018 detected with observations (a) at 1.4 GHz (SMOS) and (b) at 19 GHz (SSMIS). Seven regions are outlined.

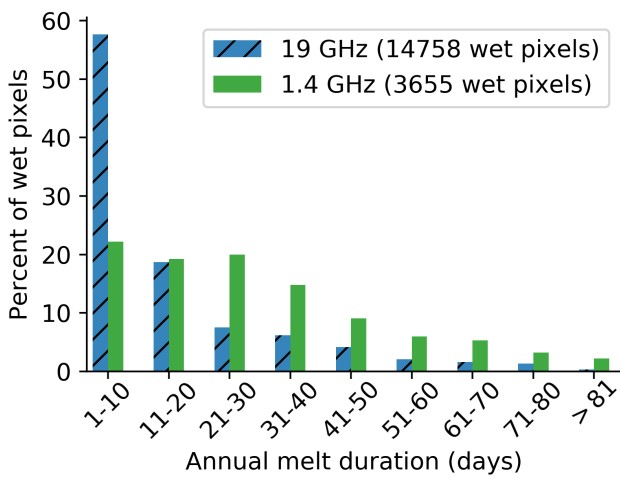

**Figure 4.** Annual melting duration distribution of wet pixels detected with 1.4 GHz (plain green) and 19 GHz (hatched blue) over the whole Antarctica for each summer season from 2010 to 2018.

variability is observed. Zheng et al. (2019) studied the Antarctic Peninsula with satellite radiometer and scatterometer as well as climate model. They found that over the period 2010-2017 the lowest wet snow extent is observed during the 2013/14 summer season, whereas the largest is observed during 2015/16. These two particular events are also retrieved by SMOS and SSMIS during this period.

Figure 5 (bottom) shows the number of days detected as melting at 1.4 GHz but dry at 19 GHz. In 2013/14, 2.6 days on average are only detected as melting by SMOS over a surface of 35,625 km$^2$ (57 pixels). In 2015/16, 12.3 days on average are only detected as melting by SMOS over a surface of 83,125 km$^2$ (133 pixels), which is 57% and 24% larger than in 2013/14 and 2014/15, respectively. As 2015/16 is known to be submitted to an intensive melting event in Antarctic Peninsula due to a

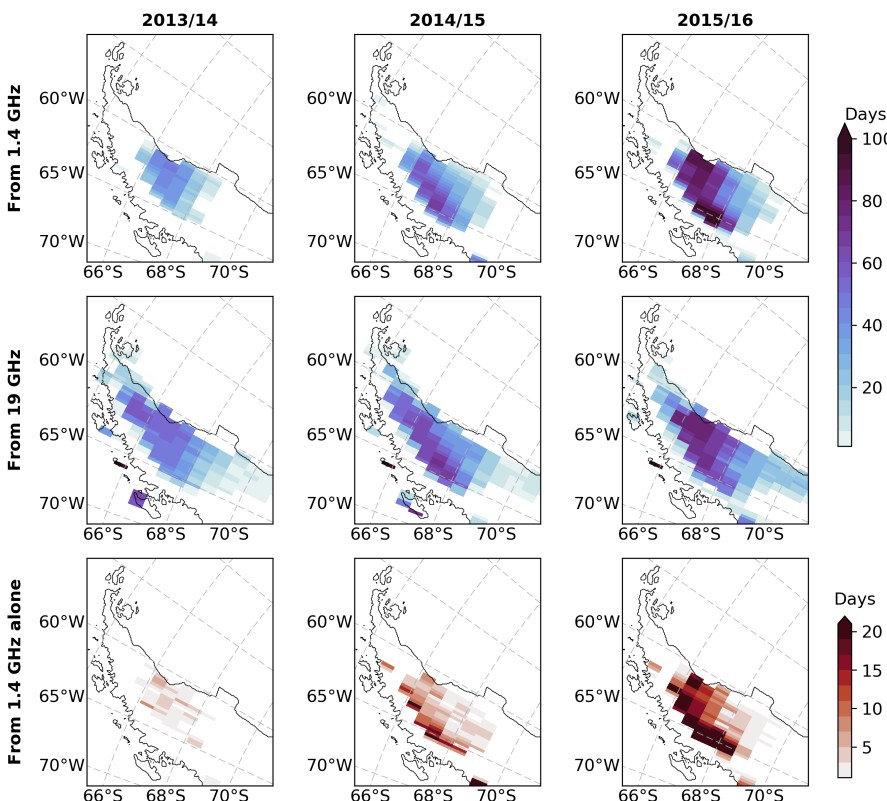

**Figure 5.** Annual melting duration (days) over the Antarctic Peninsula detected with observations (top) at 1.4 GHz and (middle) at 19 GHz from 2013/14 to 2015/16. (bottom) Number of days detected as melting at 1.4 GHz but dry at 19 GHz.

strong El-Niño event (Nicolas et al., 2017), this could suggest that 1.4 GHz provides additional information to 19 GHz in the case of intense melting events. In this way, Wiesenekker et al. (2018) showed that a stronger than normal foehn wind, which is a hot, dry wind on the downwind side of a mountain range, happens over the Peninsula in 2015/16. This generates an increasing

in melt near the foot of the Antarctic Peninsula mountains. This area matches the pixels where 1.4 GHz observations detected more than 20 days not detected by 19 GHz (Figure 5). Moreover, Datta et al. (2019) also found that high melt occurrence induced by foehn wind are observed in 2015/16, and they highlighted that this foehn wind increases the meltwater percolation up 2-m depth along the mountains. This suggests that SMOS observations could provide information about a part of snowpack in depth, which is not reached by SSMIS observations.

Figure 6 maps for the whole continent the mean number of melting days detected at 1.4 GHz without concurrent detection at 19 GHz during summer season over our dataset. It shows that the geographical distribution is related to the total number of melt event (Figure 3), meaning that all the areas are concerned by the differential detection at both frequencies. On average, 10±8 days are detected only by SMOS. Moreover, over a total of about 117,000 melting days taking all pixels and summer seasons together detected at 1.4 GHz, 28% are not concurrently detected at 19 GHz. These melting days happen on 1 February

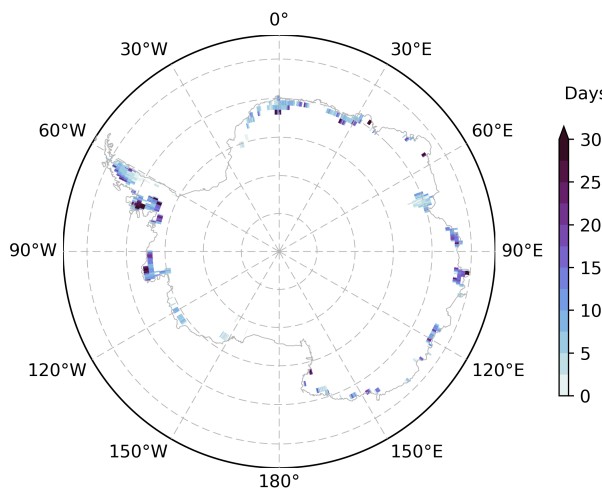

**Figure 6.** Mean melting days by summer season detected as melting at 1.4 GHz but dry at 19 GHz.

± 23 days on average, i.e. at the end of summer season. Conversely, over 225,000 melting days detected by 19 GHz during the same period, 66% are not concurrently detected at 1.4 GHz.

## 5 Sensitivity to liquid water content

The sensitivity to liquid water at 1.4 GHz is investigated in order to understand the signal variations observed in Antarctica and to investigate the observed differences with the 19 GHz melt detection.

### 5.1 Microwave emission modeling

$T_B$ is simulated with the multi-layered Dense-Medium Radiative Theory model (DMRT-ML, Picard et al., 2013), available online http://gp.snow-physics.science/dmrtml. This model is based on the radiative transfer theory (Tsang and Kong, 2001). The snowpack is represented by a stack of snow horizontal layers defined by their thickness, temperature, density, grain size, and liquid water content (LWC). Simulations are performed at 1.4 GHz and 19 GHz with an incidence angle of 55°.

A synthetic snowpack is assumed to run simulations. It has a total thickness of 1000 m, and is divided in layers of 5 cm from the surface to 500 m and 50 m below. Temperature is 273 K from the surface to 5 m depth, then constant at 263 K up to 500 m depth and finally, linearly increasing to reach 273 K at the bottom. Density linearly increases from 300 kg m⁻³ at the surface to 917 kg m⁻³ at 100 m in depth and is constant below (Leduc-Leballeur et al., 2015). Grain size is constant at 1 mm. Picard et al. (2013) showed that grain size has an effect on the sensitivity to LWC at 19 GHz. Nevertheless, it is not expected at 1.4 GHz because the wavelength is much larger than grain size and scattering by grains can be neglected (Mätzler, 1987).

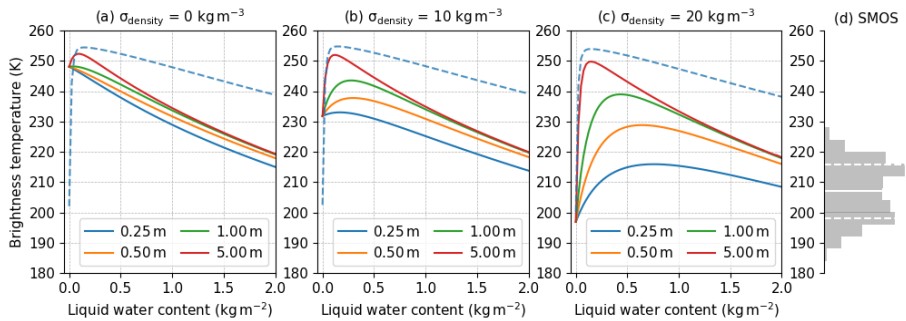

**Figure 7.** (a-c) DMRT-ML brightness temperature at H polarization (K) as a function of liquid water content for several wet snow thickness in the upper snowpack (colors) at 1.4 GHz (solid lines) and 0.25 m of wet snow at 19 GHz (dashed line) with three density variabilities ($\sigma_{density}$). (d) Daily winter SMOS observations distribution (cf. text for details) with mean (white solid) and standard deviation (white dashed).

## 5.2 Effect of snow density vertical variability

By modeling L-band emission at Dome C on the Antarctic Plateau, Leduc-Leballeur et al. (2015) highlighted that layering must be considered to obtain reliable $T_B$ estimation. To assess if this is also the case for wet snow, the simulations are performed with a smooth density profile and two density profiles with an added Gaussian noise of standard deviation of 10 kg m$^{-3}$ and 20 kg m$^{-3}$, respectively, between the surface and 300 m depth. Figure 7 shows the DMRT-ML simulations at both 1.4 GHz and 19 GHz as a function of LWC, and for various thicknesses of wet snow.

For the dry snowpack (LWC = 0 kg m$^{-2}$), the layering significantly decreases $T_B$H from 248.1 K for the smooth density profile to 231.8 K and 196.9 K for the density profiles with standard deviation 10 kg m$^{-3}$ and 20 kg m$^{-3}$, respectively. In wet snow condition, the layering effect becomes weaker as the LWC increases and is insignificant ($<$ 4 K variations) for LWC larger than 1 kg m$^{-2}$ or when water is spread over a large thickness. Thus, between dry and wet conditions, $T_B$H difference increases with the layering.

Figure 7d shows daily SMOS $T_B$H from June to August – a period when snow is expected to be always dry – in 2010-2018. The histogram only includes pixels where melting has been detected at least once and where ice thickness is 1000$\pm$50 m to match with the snowpack configuration used for simulations. The SMOS $T_B$H average is 206.9$\pm$8.9 K. This suggests that simulations with a density variability lower than 10 kg m$^{-3}$ overestimate the dry $T_B$H and thus underestimate the variations between dry and wet snow at 1.4 GHz. We thus now consider the case of a density variability of 20 kg m$^{-3}$ only.

The simulations show that $T_B$H at 1.4 GHz increases from dry to wet by 19 K when the wet snow layer is 0.25 m and 53 K when it is 5 m (Figure 7c). While in both cases, the change is high and detectable, this highlights that not only the total column amount of liquid water is important but also the distribution in depth. Additionally, Figure 7c shows that the maximum of increase of $T_B$H is reached for LWC of 0.75 kg m$^{-2}$ and 0.15 kg m$^{-2}$, respectively for the 0.25 m and 5 m thick wet snow layers. This means that the LWC sensitivity of 1.4 GHz $T_B$H is weaker when liquid water is confined in the uppermost tens of

centimeters of the snowpack. This is the rationale for choosing a lower first guess $a\sigma$ for the detection algorithm at 1.4 GHz than at the higher frequencies (Section 3).

Additionally, Figure 7c shows that regardless of the wet layer thickness, $T_BH$ reaches a maximum at a certain LWC value, which decreases when the wet layer becomes thicker. Thus, an increase in LWC is not detectable because of the $T_B$ saturation. This jeopardizes the possibility of using microwave observations to estimate LWC values or even the wet layer thickness.

By contrast, at 19 GHz, the density variability has no effect and the $T_BH$ variations are mainly driven by LWC. A sharp increase of 54 K is observed and the maximum is reach for LWC of 0.15 kg m$^{-2}$. The thickness of the wet snow layer has no effect (not shown in Figure 7c).

As a conclusion, these simulations show that 19 GHz is more sensitive to liquid water than at 1.4 GHz and that other factors such as the vertical distribution of the water or the layering have a lesser influence. This indicates that detection of melt occurrence at the surface is more robust at 19 GHz.

### 5.3   Effect of the wet snow depth

We explore here the situation when the wet snow layer is buried under a layer of dry firn. This corresponds to the end of summer when the snowpack freezes up from the surface, or on the ice shelves where melt water enters the crevasses and accumulates at depth. The simulations are performed with a wet snow layer (0.2 kg m$^{-2}$), progressively moved down from the surface to 400 m depth. The wet layer thickness is 1 m at 1.4 GHz and 0.1 m at 19 GHz to moderate the sensitivity effect presented in the previous section. Results highlight that $T_BH$ is maximum when wet snow is at the surface for both frequencies and decreases within a few meters at 19 GHz and more gradually at 1.4 GHz (Figure 8). $T_BH$ is still more than 10 K higher than in dry conditions when the wet layer is at 60 m depth at 1.4 GHz. Deeper than 100 m, the difference between dry and wet $T_BH$ is lower than 3 K, i.e. lower than the noise level with SMOS.

At 19 GHz, the simulation shows a $T_BH$ variation of 2 K between dry and wet when the wet snow is at 5 m depth. Thus, the sensitivity to liquid water is relatively quickly lost at this frequency if the water percolate deep into the firn. However, observations at 19 GHz should still be suitable for the detection of remnant liquid water at the end of the season, and when the snowpack is continuous, i.e. without crevasse.

These results suggest that despite a lower sensitivity at 1.4 GHz, liquid water could be detected with SMOS up to several tens of meters at depth and this is a new information compared to that provided by existing melt product derived from 19 GHz and higher frequencies observations. The difference observed between 19 GHz and 1.4 GHz could be exploited to determine if the melt event was limited to the few first centimeters of snowpack or if water has percolated over a sufficient thickness to be detected by SMOS.

### 6   Conclusions

The L-band brightness temperature ($T_B$) from SMOS satellite has been explored to retrieve information about the melt season in Antarctica. Daily melt occurrence can be retrieved using previously developed algorithms for higher frequencies (Zwally and

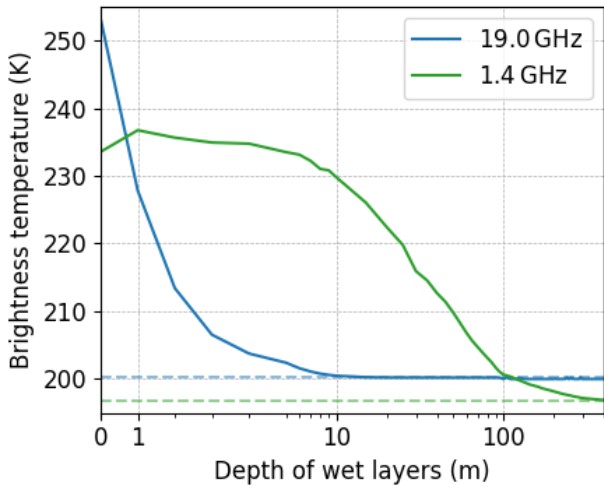

**Figure 8.** DMRT-ML brightness temperature at H polarization (K) for 55° of incidence angle as a function of the wet snow layer depth within the snowpack for a wet layer thickness of 1 m at 1.4 GHz (green) and 0.1 m at 19 GHz (blue). Values for a dry snowpack are in dashed lines.

Fiegles, 1994; Torinesi et al., 2003) after a slight adaptation to account for the lower sensitivity at 1.4 GHz. The comparison of
melt detected at 1.4 GHz and 19 GHz (Picard and Fily, 2006) shows a lower rate of detection at 1.4 GHz. In particular, SMOS misses short, probably weak, events, which are in contrast perfectly detected by SSMIS.

A theoretical analysis has been performed using a snowpack emission radiative transfer model (DMRT-ML) in order to estimate the sensitivity of $T_B$ at 1.4 GHz and 19 GHz to liquid water content (LWC) and water distribution in the snowpack. As expected from previous studies, a clear increase in $T_B$ happens when snow becomes wet. However, the simulations clearly
demonstrate that 1.4 GHz is less sensitive than 19 GHz, especially when liquid water stays within the top centimeters of the snowpack. A thick wet layer ($>$ about 0.5 m) is required to trigger a sharp and detectable $T_B$ increase. Despite this limited sensitivity, the simulations show that 1.4 GHz is suitable to detect wet snow buried under a dry surface. For instance, an increase in $T_B$ higher than 10 K with respect to a dry snowpack can be observed with liquid water at up to 60 m depth according to the simulation configuration.
An avenue is a combined use of both frequencies to determine if a melt event was limited to the surface of the snowpack or if it was intense enough to inject water at depth. However, further algorithmic work is needed to exploit this possibility of deep water detection with SMOS.

*Author contributions.* MLL, GP and GM led the study and performed the analysis. AM and YHK supported for using the SMOS observations. All authors contributed to the manuscript.

*Data availability.* Daily occurrence of melt retrieved from SMOS available at www.catds.fr/Products/Available-products-from-CEC-SM/
CryoSMOS-project

*Competing interests.* The authors declare that they have no conflict of interest.

*Acknowledgements.* This study benefited from ESA support through the CryoSMOS project (contract 4000112262/14/I-NB) and the French
space agency (CNES) support through the SMOS TOSCA project. SMOS L3 product comes from the CATDS, managed for the CNES by
IFREMER (Brest, France). We would also like to thank the Editor and the two anonymous reviewers for their very helpful comments.

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
