# Peer review of "Melt in Antarctica derived from SMOS observations at L band"

_The Cryosphere, 2019_

## Referee Comment (RC1) · Anonymous Referee #1 · 7 Oct 2019

Interesting demonstration of SMOS (1.4 GHz) capability for melt detection compare to higher frequency (19 GHz). To my knowledge this is the first time that such a comparison has been done. Even if the observed results were expected : less sensitivity at 1.4 than at 19 GHz, the differences are well described and analysed. I suggest that the authors put more emphasis on these differences that could bring complementary climatological information compared to SSMIS. In that sense, the Fig. 7 is very interesting (mean melting days detected at 1.4 GHZ but dry at 19 GHz). What are the temporal variations of such observations over the SMOS period? Do you observe particular events, for particular years? For example, the years 2002/2003 and 2015/2016 are known to be particularly wet in the Antarctic Peninsula due to a strong ENSO events. See Zheng et al. 2019 RSE, 232 : Variations in Antarctic Peninsula snow liquid water

during1999–2017 revealed by merging radiometer, scatterometer and model estimations

This is unfortunate that the Fig. 1 stops in April 2015, because 2016 could be a good example of differences between 1.4 and 19 GHz data?

See also Wiesenekker et al., 2018. A Multidecadal Analysis of Föhn Winds over Larsen C Ice Shelf from a Combination of Observations and Modeling. Atmosphere 9(5), 172. https://doi.org/10.3390/atmos9050172. for the relationship between particular Föhn events and melting.

I also suggest to add Zheng et al. 2019 reference (and others) for mentioning scatterometer and radar capabilities compared to radiometers (not mentioned in the paper).

The DMRT-ML analysis is a very good added-value to this paper.

Also, could you specify which ice/water mask do you used for SMOS? same as for resampled SSMI mask? source of error? Does the Fig. 5 cover the entire SMOS period and for the whole Antarctica?

---

## Referee Comment (RC2) · Anonymous Referee #2 · 18 Oct 2019

The authors have performed a study to detect the melt occurrence in Antarctica using SMOS observations. Authors have compared the SMOS detection results to those obtained using 19.7 GHz passive observations. This study provides very good results – showing the usefulness of SMOS observations for melt occurrence detection. Theoretical analysis explains well the differences between the L-band and 19.7 GHz observations and provided very nice basis on understanding the importance on having observations at both frequencies to better monitor the melt occurrences. The manuscript is well written and structured, it is easy to read and understand. The aim of the study is clearly explained, and conclusions are well provided. Scientifically, the paper is solid, it provides interesting and important new information on how to better detect and monitor the ice melt on Antarctica. I recommend this paper to be published and have only some

minor comments to be considered before publishing: The comments are listed below.

1) Line 46: I assume the authors are using CATDS data from 50 to 55 degrees.

2) Lines 120-121: The selected temperature profile is a little strange: From surface to 10 m: 273 K, then constant 263K to 500m depth. Are the authors really using this, or should it be from surface to 10 m dropping from 273K to 263 K?

3) Figure 5: Based on the model results, the selected density profile has a large impact. Tb as a function of the liquid water content is totally different if a smooth density profile is applied. Daily winter SMOS observations are compatible with the third density profile (20 kg/m3). How much the density profile varies in real life, may there be an additional source of uncertainty for the SMOS based estimations?

4) Line 138: Odd sentence, maybe "have been selected" should not be there.

5) Line 162: Maybe, to clarify the readers, the authors could use: "The wet layer thickness" instead of "The layer thickness"

6) Line 174: The sentence is a bit confusing starting from words "or if the event was produce a lot . . ."

7) Figure 6: The caption text is not as informative as it could be. "as a function of the wet snow depth" => how about: "as a function of the wet snow layer depth". By adding word layer, it is easier to understand that the simulation is done using constant layer thickness but in different depths. Also, consider adding the layer thicknesses here.

---

## Author Comment (AC1) · 13 Dec 2019

Thanks to devote time to the review of our manuscript. The manuscript has been revised following your comments and suggestions. In particular, we enhanced the comparison between 1.4 GHz and 19 GHz to highlight their complementary climatological information. We answered your comments (introduced by '»') in the following.

»Interesting demonstration of SMOS (1.4 GHz) capability for melt detection compare to higher frequency (19 GHz). To my knowledge this is the first time that such a comparison has been done. Even if the observed results were expected : less sensitivity at 1.4 than at 19 GHz, the differences are well described and analysed. I suggest that the authors put more emphasis on these differences that could bring complementary climatological information compared to SSMIS. In that sense, the Fig. 7 is very interesting (mean melting days detected at 1.4 GHz but dry at 19 GHz). What are the temporal variations of such observations over the SMOS period? Do you observe particular events, for particular years? For example, the years 2002/2003 and 2015/2016 are known to be particularly wet in the Antarctic Peninsula due to a strong ENSO events. See Zheng et al. 2019 RSE, 232 : Variations in Antarctic Peninsula snow liquid water during 1999–2017 revealed by merging radiometer, scatterometer and model estimations. This is unfortunate that the Fig. 1 stops in April 2015, because 2016 could be a good example of differences between 1.4 and 19 GHz data? See also Wiesenekker et al., 2018. A Multidecadal Analysis of Föhn Winds over Larsen C Ice Shelf from a Combination of Observations and Modeling. Atmosphere 9(5), 172. https://doi.org/10.3390/atmos9050172 for the relationship between particular Föhn events and melting.

In order to improve the comparison between 1.4 GHz and 19 GHz, we extended the Section 4 with a more detailed view of the case, when day is detected as melting by SMOS but dry by SSMI. We mainly based this analysis on the articles that you suggest and added a focus on the Antarctic Peninsula. We focused the analysis on the period 2013-2013 2013-2016 when the variation are stronger. Fig. 1 have been extended over this period, and we added a new figure (Fig 1r) with some Peninsula maps to better highlighted the temporal variation. Thus, we added the following text in the end of the Section 4 and putted the figure previously named 'Figure 7' in the following of this Section:

However, it also happens that some melting days are detected with the 1.4 GHz observations but not with the 19 GHz observations. This case is illustrated with the example of the Antarctic Peninsula provided by Figure 1r for the three summer seasons from 2013 to 2016. This area is known to be submitted each year to a long melting season, but an interannual variability is observed. Zheng et al. (2019) studied the Antarctic Peninsula with satellite radiometer and scatterometer as well as climate model. They

Interactive
comment

found that over the period 2010-2017 the lower wet snow extend is observed in during the 2013/14 summer season, whereas the largest is observed during 2015/16. These minimum and maximum are also retrieved by SMOS and SSMI during this period. Figure 1r (bottom) shows the number of days detected as melting at 1.4 GHz but dry at 19 GHz. In 2013/14, 2.6 days on average are only detected as melting by SMOS over a surface of 35,625 km2 (57 pixels). In 2015/16, 12.3 days on average are only detected as melting by SMOS over a surface of 83,125 km2 (133 pixels), which is 57% and 24% larger than in 2013/14 and 2014/15, respectively. As 2015/16 is known to be submitted to an intensive melting event in Antarctic Peninsula due to a strong El-Nino event (Nicolas et al., 2017), this could suggest that 1.4 GHz provide another information than 19 GHz in the case of intense melting events. In this way, Wiesenekker et al. (2018) showed that a stronger than normal foehn wind, which is a hot, dry wind on the downwind side of a mountain range, happens over the Peninsula in 2015/16. This generates an increasing in melt near the foot of the Antarctic Peninsula mountains. This area matches the pixels where 1.4 GHz observations detected more than 20 days not detected by 19 GHz (Figure 1r). Moreover, Datta et al. (2019) also found that high melt occurrence induced by foehn wind are observed in 2015/16, and they highlighted that the foehn wind increases the meltwater percolation up 2-m depth along the mountains. This suggests that SMOS observations could provide information about a part of snowpack in depth, which is not reaches by SSMI observations.

Figure 7 (now 6) maps for the whole continent the mean number of melting days detected at 1.4 GHz without concurrent detection at 19 GHz during summer season over our dataset. It shows that the geographical distribution is related to the total number of melt event (Figure 3), meaning that all the areas are concerned by the differential detection at both frequencies. On average, $10\pm8$ days are detected only by SMOS. Moreover, over a total of about 117,000 melting days taking all pixels and summer seasons together detected at 1.4 GHz, 28% are not concurrently detected at 19 GHz. These melting days happen on 1 February $\pm$ 23 days on average, i.e. at the end of summer season. Conversely, over 225,000 melting days detected by 19 GHz during

the same period, 66% are not concurrently detected at 1.4 GHz.

Figure 1r: Annual melting duration (days) over the Antarctic Peninsula detected with observations (top) at 1.4 GHz and (middle) at 19 GHz from 2013/14 to 2015/16. (bottom) Number of days detected as melting at 1.4 GHz but dry at 19 GHz.

»I also suggest to add Zheng et al. 2019 reference (and others) for mentioning scatterometer and radar capabilities compared to radiometers (not mentioned in the paper).

As you suggest in order to improve the context description in Introduction, we added sentences and provided references including Zheng et al. (2019) to highlight the capability of active sensors to detect melt on the ice sheet. We added in the text: "Various detection algorithms have been developed for active sensors (e.g. Nghiem et al., 2001, 2005; Ashcraft and Long, 2006; Kunz and Long, 2006; Hall et al., 2009; Trusel et al., 2012; Zheng et al., 2019) and passive sensors (e.g. Mote et al., 1993; Ridley, 1993; Zwally and Fiegles, 1994; Abdalati and Steffen, 1997; Torinesi et al., 2003; Liu et al., 2005, 2006; Tedesco, 2007; Tedesco et al., 2007) and applied in the Greenland and Antarctica ice sheets.".

»The DMRT-ML analysis is a very good added-value to this paper.

»Also, could you specify which ice/water mask do you used for SMOS? same as for resampled SSMI mask? source of error?

The mask used here is the mask associated to the EASEGrid 2.0 map projections. It is available on the NSIDC website: https://nsidc.org/data/nsidc-0609. Brodzik et al. (2011) derived this Land-Ocean-Coastline-Ice (LOCI) classification from the MODIS land cover product. We added this information to the SMOS observations description in Section 2.1.

As SMOS and SSMI datasets are not built in the same grid some collocation error can happen. We added a description of the used method to compare the two datasets in Section 2.2: "To compare SMOS and SSMI datasets, the SSMI observations and

products are collocated within the SMOS grid using the nearest neighbour method. If the nearest neighbour is not flagged as 'land' in the SSMI grid, the pixel was removed from our analysis to avoid the error of comparison between the two frequencies. In this way, about 50 pixels are excluded, which doesn't affect the statistical significance of the comparison results."

Note that the development of a Level 3 SMOS product within a polar stereographic projection is in progress by CATDS team, but up to now the official release is available from the February 2018 to present and the whole timeseries from 2010 is not yet ready.

»Does the Fig. 5 cover the entire SMOS period and for the whole Antarctica?

Fig. 5a-c refers to the DMRT-ML simulations. On Fig. 5d, the histogram only includes SMOS pixels fulfilling the two conditions: 1) have been detected as 'melting' at least once over the period 2010-2018, and 2) the ice thickness is $1000\pm50$Ăm. This is described in Section 5.2 at the beginning of the third paragraph. We added a cross-reference to text in the figure legend to find more information.

––––––––––––––––––––––

[Figure]

**Fig. 1.** Figure 1r: Annual melting duration (days) over the Antarctic Peninsula detected with observations (top) at 1.4 GHz and (middle) at 19 GHz from 2013/14 to 2015/16. (bottom) Number of days detected as m

---

## Author Comment (AC2) · 13 Dec 2019

We appreciated very much your comments and efforts in reviewing this paper. The manuscript has been revised, according to comments and suggestions provided. We answered your comments (introduced by '»') in the following.

»The authors have performed a study to detect the melt occurrence in Antarctica using SMOS observations. Authors have compared the SMOS detection results to those obtained using 19.7 GHz passive observations. This study provides very good results –showing the usefulness of SMOS observations for melt occurrence detection. Theoretical analysis explains well the differences between the L-band and 19.7 GHz observations and provided very nice basis on understanding the importance on having ob-

servations at both frequencies to better monitor the melt occurrences. The manuscript is well written and structured, it is easy to read and understand. The aim of the study is clearly explained, and conclusions are well provided. Scientifically, the paper is solid, it provides interesting and important new information on how to better detect and monitor the ice melt on Antarctica. I recommend this paper to be published and have only some minor comments to be considered before publishing.

»The comments are listed below.

1) Line 46: I assume the authors are using CATDS data from 50 to 55 degrees.

Yes. We now specified in the text: "TB at vertical (V) and horizontal (H) polarizations for the 50-55° average range of incidence angle are used here.".

2) Lines 120-121: The selected temperature profile is a little strange: From surface to 10 m: 273 K, then constant 263K to 500m depth. Are the authors really using this, or should it be from surface to 10 m dropping from 273K to 263 K?

There is a mistake in the text for the temperature profile description. In fact, the used profile is: from the surface to 5 m is 273 K, then constant 263 K to 500 m. We choose to fix the temperature at 273 K within the first 5 m in order to limit the temperature variations effect and highlight the LWC effect.

3) Figure 5: Based on the model results, the selected density profile has a large impact. Tb as a function of the liquid water content is totally different if a smooth density profile is applied. Daily winter SMOS observations are compatible with the third density profile (20 kg/m3). How much the density profile varies in real life, may there be an additional source of uncertainty for the SMOS based estimations?

It is really difficult to have a reliable estimation of the density variability range, due to the lake of in situ measurement and the large penetration depth of SMOS. For example, at Dome C, we estimated the density variability about 25-30Ăkg/m3 close to the surface (Leduc-Leballeur et al., 2015). However, the snowpack structure in Dome C area is

typical of the dry snow region, which is completely different of the wet snow area. Here, thanks to the simulations with 3 values of density variability and the comparison with the SMOS observations during winter, we can suggest that a variability lower than 10 kg/m3 is not very probable and 20 kg/m3 was selected (Figure 5). However, the standard deviation of the SMOS histogram (206.9±8.9 K) also suggests a variability which could be in part linked to a change in the density variability of the profile. So, as you highlight, the lake of knowledge of the density variability can adding uncertainty to simulate the SMOS observations.

4) Line 138: Odd sentence, maybe "have been selected" should not be there.

Thank to have notice that. We removed "have been selected".

5) Line 162: Maybe, to clarify the readers, the authors could use: "The wet layer thickness" instead of "The layer thickness"

We added "wet" to clarify the sentence.

6) Line 174: The sentence is a bit confusing starting from words "or if the event was produce a lot..." To clarify, we corrected this sentence part as:

"if water has percolated over a sufficient thickness to be detected by SMOS.".

7) Figure 6: The caption text is not as informative as it could be. "as a function of the wet snow depth" => how about: "as a function of the wet snow layer depth". By adding word layer, it is easier to understand that the simulation is done using constant layer thickness but in different depths. Also, consider adding the layer thicknesses here.

We changed the caption text for Figure 6 as: "DMRT-ML brightness temperature at H polarization (K) for 55deg of incidence angle as a function of the wet snow layer depth within the snowpack for a wet layer thickness of 1 m at 1.4 GHz (green) and 0.1 m at 19 GHz (blue). "
* * *